# Knowledge Domain and Emerging Trends in Vinegar Research: A Bibliometric Review of the Literature from WoSCC

**DOI:** 10.3390/foods9020166

**Published:** 2020-02-10

**Authors:** Xiang-Long Zhang, Yu Zheng, Meng-Lei Xia, Ya-Nan Wu, Xiao-Jing Liu, San-Kuan Xie, Yan-Fang Wu, Min Wang

**Affiliations:** State Key Laboratory of Food Nutrition and Safety, Key Laboratory of Industrial Fermentation Microbiology, Ministry of Education, College of Biotechnology, Tianjin University of Science & Technology, Tianjin 300457, China; xl_zhang@mail.tust.edu.cn (X.-L.Z.); yuzheng@tust.edu.cn (Y.Z.); mlxia@tust.edu.cn (M.-L.X.); 18820944@mail.tust.edu.cn (Y.-N.W.); lxj9704@163.com (X.-J.L.); xskuan@163.com (S.-K.X.); amwyvonne@163.com (Y.-F.W.)

**Keywords:** vinegar, intellectual structure, knowledge mapping, CiteSpace, bibliometrics

## Abstract

Vinegar is one of the most widely used acidic condiments. In recent decades, rapid advances have been made in the area of vinegar research, and the intellectual structure pertaining to this domain has significantly evolved. Thus, it is important that scientists keep abreast of associated developments to ensure an appropriate understanding of this field. To facilitate this current study, a bibliometric analysis method was adopted to visualize the knowledge map of vinegar research based on literature data retrieved from the Web of Science Core Collection (WoSCC) database. In total, 883 original research and review articles from between 1998 and 2019 with 19,663 references were analyzed by CiteSpace. Both a macroscopical sketch and microscopical characterization of the whole knowledge domain were realized. According to the research contents, the main themes that underlie vinegar research can be divided into six categories, that is, microorganisms, substances, health functions, production technologies, adjuvant medicines, and vinegar residues. In addition to the latter analysis, emerging trends and future research foci were predicted. Finally, the evolutionary stage of vinegar research was discerned according to Shneider’s four-stage theory. This review will help scientists to discern the dynamic evolution of vinegar research, as well as highlight areas for future research.

## 1. Introduction

Vinegar is one of the most familiar acidic condiments around the world [1]. It has been widely used in almost all civilizations since ancient times and plays important roles as a food-flavoring, medicinal, preservative, and cleaning agent. Nonetheless, very little is known in relation to the origin of this condiment [2]. Regardless of the variations associated with vinegar from disparate regions and eras, in essence, vinegar is a dilute solution of acetic acid derived from the oxidation of ethanol following microbial fermentation rather than chemical synthesis [3]. The Food and Agriculture Organization of the United Nations (FAO) and the World Health Organization (WHO) define vinegar as an edible liquid exclusively produced from starchy and/or sugary raw materials by two sequential processes including alcohol and acetic acid fermentation [4]. Throughout the world, vinegar is traditionally brewed initially from fruits, cereals, vegetables, animal products (e.g., honey and whey) or alcohol, and vinegar production strategies can be divided into solid-state fermentation (SSF) and liquid-state fermentation (LSF) methods (Table 1, Figure 1). SSF technology is predominantly used in Asia, especially in China. Conversely, LSF, including surface static fermentation and submerged fermentation, is predominantly used in European countries to manufacture traditional vinegar and accelerate industrial production. As for the composition of vinegar, it contains a variety of entities including organics, inorganics, polymers, (nano)particles, macromolecules, small molecules, and polar and nonpolar molecules, as well as constant and trace substances [5,6,7,8,9]. In addition to its role as a condiment, vinegar also exhibits many health benefits including antimicrobial, antioxidant, cholesterol-lowering, antiobesity, antihypertensive, and immunostimulatory effects [1,10,11].

Worldwide, the development of the vinegar industry is accelerating and improving all the time. With the rapid growth of this area, mapping the knowledge domain of vinegar research is becoming increasingly important as scientists attempt to attain a holistic understanding of this field while keeping abreast of important developments. As an important form of written research records, scientific literature clearly reflects the structure and dynamics of knowledge domains [12]. However, the generation of large-scale literature reviews from extremely large corpora of associated research papers represents a challenge for scientists and researchers. The study of traditional reviews on vinegar research has conceivably become an insufficient means of developing a comprehensive understanding in this field. Following advances in modern information technology and statistics, bibliometric analyses, which were regarded as powerful tools for both the summarization of historical research achievements and predictors of future research trends using scientific literature databases, have become increasingly popular in academic circles for the generation of reviews in many research fields such as regenerative medicine [13] and food toxicology [14]. However, a specific bibliometric analysis of vinegar research has not yet been performed.

CiteSpace is one of the most popular bibliometric analysis softwares used for the statistical visualization of scientific literature across the globe. It was pioneered in early 2004 and is written in Java language [15]. Thus far, the number of total downloads for each version of this software has exceeded 130,000 (http://cluster.cis.drexel.edu/~cchen/citespace/download/). CiteSpace has three fundamental concepts: heterogeneous networks, betweenness centrality, and burst detection. Co-citation analysis is the core function of this application. CiteSpace can be used to explore the dynamic evolution of disciplinary development through time mapping from the research frontier to the knowledge base [12].

The main objectives of this study were to summarize the current status of and analyze future developing trends in the knowledge domain of vinegar research. In this current study, we used the Web of Science Core Collection database (WoSCC) to perform a simple statistical analysis of literature related to vinegar research; we utilized CiteSpace to perform a bibliometric visualization analysis. The results of this study will help scientists to better understand the development of research in this area, the associated research hot spots, and potential future research directions for vinegar research.

## 2. Materials and Methods

### 2.1. Data Acquisition

The WoSCC database has been used in a large number of bibliometric studies and is set up for this type of analysis. Thus, all reference data used in our study were retrieved from the WoSCC including Science Citation Index Expanded (SCIE) studies of Clarivate Analytics. The first article pertaining to vinegar research that can be retrieved from WoSCC was published by Anklam et al. in 1998 [16]. Thus, the timespan for data retrieval was 1998 to 2019. An initial title search for ‘vinegar’ and a topic search that excluded ‘wood vinegar’ (a kind of pyroligneous liquor collected during wood carbonization [17]) and ‘vinegar fly’ (a kind of insect belonging to the genus *Drosophila*, https://www.britannica.com/animal/vinegar-fly) resulted in 1039 records (retrieval date 15 July 2019). After filtering out less representative record types, such as proceeding papers and meeting abstracts, the dataset was reduced to 883 original research articles and review articles. Full records for these 883 articles, including titles, authors, abstracts, and cited references (19,663), were exported in the form of plain text from WoSCC and were used as the dataset (see Appendix A) for this study.

### 2.2. Visualization and Analysis

In this study, version 5.4.R1 of CiteSpace (http://cluster.cis.drexel.edu/~cchen/citespace/download/) was used for all visual analyses. After data acquisition, the dataset was exported to CiteSpace for further analysis [12]. The following is an outline of the process that was followed for this review. Firstly, appropriate disciplines in vinegar research were visualized by CiteSpace using a dual-map overlay. Secondly, statistical analysis and comparison of literature outputs were performed based on the journal citation reports (JCRs) at WoS and co-occurrence analysis in CiteSpace. Both the analysis of output trends and the comparison of research outputs between different journals were performed according to JCR, while the co-occurrence analysis of the subject categories and collaborating analyses between countries and institutions were performed using CiteSpace. Thirdly, the intellectual structure of vinegar research was visualized through co-citation analysis, and finally, the emerging trends and outlook were discerned via burst detection and embroiling insights from other research areas. For all network visualizations, time slicing was set to one-year. If not specified, the top 50 levels of the most prevalent items from each slice were selected, and no pruning algorithm was applied. CiteSpace default values were used for all of the other requisite parameters. More detailed software utilization skills can be found in the CiteSpace manual [18].

## 3. Results and Discussion

### 3.1. Disciplines and Topics Involved in Vinegar Research

A dual-map overlay of the literature on vinegar research represents the entire dataset in the context of a global map of science generated from 10,000 journals indexed in the WoS [19]. In a dual-map, both citing (the left map) and cited (the right map) maps are shown, and an overlay of the given dataset facilitates visualization of the disciplinary concentrations of these articles and how they connect other regions in the global map through their citation links [20].

Figure 2 shows a dual-map overlay of acquired vinegar articles published between 1998 and 2019. All coloured curves originating from the citing map and pointing to the cited map represent the paths of the citation links. Both citing and cited maps are divided into thematic areas based on publishing journals, and each area is labeled with the most common words in the titles of corresponding journals [20]. Labels in the adjacency of the launching areas display corresponding disciplines in which citing articles were published [19]. As shown in Figure 2, the literature on vinegar research appears in several areas. The area in purple at the top labeled physics/materials/chemistry; the area in blue near the top labeled ecology/earth/marine; the area in yellow in the upper middle section labeled veterinary/animal/science; the areas in orange in the middle labeled molecular/biology/immunology; and green in the lower left labeled medicine/medical/clinical. However, all these citing articles were concentrated in the veterinary/animal/science and molecular/biology/immunology disciplines. Furthermore, all the cited articles were concentrated in environmental-/toxicology-/nutrition-, chemistry-/materials-/physics-, and molecular-/biology-/genetics-related journals.

The bar charts shown on the citing map depict stepwise drifts in trajectories from the citing behavior of related articles. The first bar on the left of the bar chart represents the amount of shift in 1999 with reference to the weight center of the disciplines involved in 1998 [19]. The charts show that the distance of the shifts increased substantially in 2000, 2002, and 2005–2006. This phenomenon can also be observed in the partial enlargement of the citing trajectory in the upper half of Figure 2. This indicates that, in these years, new papers may have been published in different scientific disciplines compared with the preceding years. The starting position of the citing trajectory is predominated by publications in the discipline of veterinary/animal/science, while in the drift years, the trajectory appeared to be influenced by activities in regions near to the disciplines of molecular/biology/immunology. Similarly, the cited trajectory represents the change of disciplines for cited articles. This change was predominantly influenced by activities in areas adjoining the disciplines of environmental/toxicology/nutrition and molecular/biology/genetics.

The topics associated with vinegar research can be delineated in terms of the keywords in each article in the dataset [13]. A minimum spanning tree pruning algorithm (pruning the merged network) was used to depict the network of keywords (Figure 3). Neighbor keywords are often assigned to the same articles. For example, gradient gel electrophoresis, lactic acid bacteria (LAB), and diversity are near to each other on the top of the network diagram; quality, polyphenol/phenolic compound, and antioxidant/antioxidant activity are near to each other on the lower right of the figure. According to the semantemes, all keywords can be divided into two types:

What: The subject of a study and a phenomenon of a specialty. For example, Shanxi aged vinegar (SAV), volatile compound, and acetic acid bacteria (AAB).

How: Methodologies, procedures, tools, and techniques. For example, chemometrics, solid-phase microextraction, electronic nose, and mass spectroscopy.

### 3.2. Trends and Comparison of Literature Outputs for Vinegar Research

#### 3.2.1. Trends of Research Outputs and Citations

The trace of published articles shows the development speed and progress of research, while also reflecting the concentration of research in a certain field [21]. As shown in Figure 4, although some fluctuations were observed, the number of papers published increased over time. The number of published papers rose from 19 in 1998 to 97 in 2018. It should be noted that the retrieved number of published papers in 2019 was only 44; this is because records for only half of a year were available by the end of the retrieval date.

According to the holistic development trend, the studied period can be divided into three stages. The first stage can be regarded as initiating from 1998 to 2007, with research work still in its infancy and reduced in its prevalence. A total of 179 papers were published in this 10-year period; this accounted for 20.27% of the total number of papers. The average annual publication was approximately 18 papers for this knowledge base forging period. The second stage was from 2008 to 2012. Although more papers were published in these five years with a total of 220 publications, this stage was relatively flat. On average, 44 papers were published annually; this was twice as much as that observed for the first stage. The third stage was from 2013 to 2019 (the first half-year of 2019). There was a great increase in both the number and growth rate of publications. In the 6.5 years, 484 papers were published, accounting for 54.81% of the total number of papers published; during this period, an average of nearly 75 papers were published annually. The number of publications increased at every stage of the studied period. This suggests that vinegar research has attracted the attention of more researchers during this period.

In the 22 years from 1998 to 2019, the total number of citations was 11,809, and each article was cited approximately 14 times. The research outputs were greatly improved, with the total number of citations rapidly growing. The increasing trend for the annual citation number was similar to that for the annual publication number. This also indicates that scientists tend to cite the most recently published articles [22].

#### 3.2.2. Comparison of Research Outputs between Different Subject Categories, Countries, Institutions, and Journals

In CiteSpace, category, country, and institution were selected as nodes to analyze subject categories and research work forces. Two comprehensive networks consisting of nodes representing co-occurrence subject categories, as well as collaborating countries and institutions, are mapped in Appendix A, respectively. The top ten subject categories, countries, and institutions in terms of publications in 883 related articles are shown in Table 2. The top ten journals (journal information was acquired from the retrieval result of JCR at WoS) that published articles on vinegar research are shown in Table 3.

Each node in the relation network of the subject category represents a category involved in vinegar research. The area of each node is proportional to the co-occurrence frequency of the respective subject category. The network consists of 47 nodes and 182 links (Appendix A); this indicates that a total of 47 subject categories are involved in this research field. The most common category is Food Science & Technology, which is the largest circle with a frequency of 445, followed by Chemistry (247); Chemistry, Applied (123); Biotechnology & Applied Microbiology (113); and Agriculture (96) (Appendix A, Table 2). Subject categories including Chemistry, Analytical, and Biochemical Research Methods, where some of the associated citation rings are illustrated in red (nodes are not shown in Appendix A because of their small sizes), represent research areas where the number of articles has increased rapidly. Although subject categories including Immunology, Endocrinology & Metabolism, and Oncology are much smaller, they are marked for reference herein.

At the national level, China was the largest contributor in terms of numbers of publications with 261 papers, followed by Japan, Spain, Italy, and South Korea, with 109, 101, 83, and 72 papers, respectively. The top five countries in terms of centrality (purple round in Appendix A) were China (0.57), Italy (0.54), Japan (0.48), Spain (0.31), and USA (0.31). An analysis in terms of publication and centrality indicated that China, Italy, Spain, USA, and South Korea were the main work forces in vinegar research.

The number of papers published by different research institutes was sorted. As the only institute in Italy, the University of Modena and Reggio Emilia was the most productive with 35 publications. The University of Seville and the University of Cadiz in Spain were in second and third place, with 27 and 25 papers, respectively. All top ten institutions were from Italy, China, and Spain.

Finally, the research outputs were arranged according to their source journals (Table 3). According to the rankings, *Food Chemistry* and *Journal of Agricultural and Food Chemistry* ranked first and second, with 39 and 34 articles, respectively. These two journals represent the journals with the highest impact in the respective categories (see journal key indicators in Table 3). The total number of articles published in the ten journals was 206, which accounted for 23.33% of the total number of publications. A large number of articles were also published in other journals, such as *LWT-Food Science and Technology*, *Journal of Food Science*, and *Journal of Functional Foods*. This represents important submission-related information for new researchers. Because of the different scopes and topics of the journals, we can conclude that vinegar research is interdisciplinary.

### 3.3. The Intellectual Structure of Vinegar Research

Scientific literature is an interrelated and constantly extending system, in which the intellectual structure of a research field can be articulated. It has been suggested that a better understanding of a specific topic often relies on an understanding of its relatedness to other topics [18]. Thus, the mutual citation of scientific literature reflects the structure and dynamics of a knowledge domain [14]. Citing articles and cited articles represent the temporal research front and intellectual base, respectively [23].

The co-citation relationship refers to the co-occurrence of two articles in the reference list of a paper, and the process of mining the co-citation relationship in a dataset is regarded as a co-citation analysis [12]. In CiteSpace, the co-citation relationship among different articles was visualized in a co-citation network [13].

The clustering function in CiteSpace divides the whole network into several clusters and extracts noun phrases from the titles, keywords, or abstracts of articles that cited the particular cluster. Each cluster is a group of closely coupled documents, representing different research directions and backgrounds in the field. In a cluster, each node represents a reference and is connected by lines. The size of a node indicates how many citations the reference received, and the citation tree-rings show the citation times across the timespan. The colours of co-citation lines show when a connection was made for the first time (blue for the oldest and yellow for the most recent citations). The betweenness centrality scores of these nodes indicate the transformative potential of specific articles. The nodes with high betweenness centrality are displayed in purple rings. Such nodes tend to bridge different stages of the development of a field. Citation rings in red indicate citation bursts; this means that the publication evidently has attracted great attention from the scientific community. Citation bursts provide a meaningful indicator to trace the footprint of the research focus [13,18].

#### 3.3.1. Research Clusters

Figure 5 shows a landscape network of a co-citation analysis for the top 60 references with the most annual citations. In total, 673 nodes with 2999 connections and 71 clusters were obtained. The modularity Q of the network was 0.7243, which is relatively high, suggesting that the specialties in the field are clearly defined. In the present review, all cluster labels were extracted from titles of citing articles using a log-likelihood ratio algorithm. Table 4 lists 12 clusters with more than 10 members in each cluster. Apart from Clusters #1, #3, and #4, most of the clusters in Table 4 are highly homogeneous with silhouette values above 0.7, suggesting a reliable clustering result. Besides, the mean year of a cluster indicates that Clusters #4, #9, and #12 were newly formed, while Cluster #11 was the oldest cluster.

Figure 6 shows a timeline view of how the network is divided into distinct co-citation clusters. Vertically, the clusters are arranged in descending order based on their sizes. It is evident that Clusters #0–#6 have a high concentration of nodes with citation bursts. Clusters #11, #15, and #16 do not appear to have many recent publications. Clusters #9 and #12 appear to be separate from other clusters with infrequent citation links connecting to other clusters. Cluster #0 has a sustained period of about 20 years from 1997 to 2013, whereas Cluster #15 is short-lived with an associated period of 4 years from 2004 to 2007. Clusters #1, #3, #4, and # 5 remained active until 5 years ago.

Clusters #0–#7 were deemed conventional clusters because of their larger sizes, increased sustainability, or higher concentration of citation bursts compared with other clusters. Analysis of these typical clusters can help us to understand the major specialties associated with vinegar research and their dynamic evolution. In the following discussion, we will particularly focus on the following clusters.

Cluster #(1 and 6)—*Fermented beverage & Healthy subject.* The cluster label of Cluster #1 (*Fermented beverage*) seems to indicate that the specialty of this cluster is related to vinegar beverages. However, a detailed look at the cited members in this cluster reveals that the major content pertains to the evaluation of vinegar function, which is actually close to the specialty associated with Cluster #6. Thus, these two clusters were subsequently integrated into one cluster. The split of these two similar clusters indicates that the clustering function of CiteSpace still needs to be optimized [18]. This integrated cluster subsequently became the largest cluster with 111 references. This cluster primarily focuses on studies investigating the health functions of vinegar. The evolution of this specialty can be divided into two periods. The first period is from 1997 to 1999. During this period, the health function of vinegar was first introduced to scientific research. The first article in this field was published by Liljeberg & Björck [24]. In this article, the postprandial hypoglycemic effect of vinegar was evaluated in vivo. The second period is from 2000 to 2018, with lots of burst articles published. The outstanding references in this integrated cluster are shown in Table 5.

Cluster #(3 and 4)—*New sherry & Bioactive compound.* This integrated cluster is the second largest cluster. The duration of this cluster ranges from 2003 through 2018. The main topic involved in the cluster is bioactive compounds in vinegar. Many members of this cluster refer to the bioactive compounds of phenolics. The development of this specialty can be divided into two periods. A timespan of four years from 2003 to 2006 represents the first period. In this period, the first article relating to the study of phenolics in vinegar was by Alonso et al. [25]. In this article, both the total polyphenolic content and polyphenolic compounds in Sherry vinegar were studied. It was found that there was a close correlation between the antioxidant power and the total phenolic content. The second period from 2007 to 2018 was a vibrant period in this field. The phenolic compounds in different vinegars such as traditional balsamic vinegar (TBV) [26], persimmon vinegar [27], strawberry vinegar [28], and SAV [29,30] were also studied. In addition, other bioactive compounds including ligustrazine [31] and 5-hydroxy-4-phenyl-butenolide [32] in cereal vinegar, caffeoylsophorose in purple sweet potato vinegar [33], tryptophol in black soybean vinegar [34], amino acids [35], vitamins [36], melanoidins [37,38,39], polysaccharides [6,40], and anthocyanin [41] were also studied.

Cluster #0—*Multivariate calibration*. Cluster #0 is the third largest cluster with 83 references (Table 4). The primary focus of this cluster is mainly on the analysis of substance composition of vinegar. Core contents in this cluster refer to the establishment of detection methods, especially rapid detection methods, and data processing. The timeline visualization reveals three periods of development (Figure 6). The first period was from 1997 to 2001. This period was relatively inactive without prolific publications in terms of citation counts or bursts. An inspirational article by Chiavaro, Caligiani & Palla in 1998 [42] inspired several studies in the second period. In this latter article, changes in the composition of balsamic vinegar as a function of aging were analyzed by gas chromatography-mass spectrometry (GC-MS), and chiral indicators were identified. During the second period from 2002 to 2009, high-profile articles were published. Numerous detection methods were developed and an abundance of data was generated. The most cited members of this cluster include multiple detection methods such as GC/GC-MS [43,44,45,46], nuclear magnetic resonance (NMR) [47], and near infrared (NIR) [48] spectroscopy. The third period was from 2010 to 2013, with no burst articles detected. The evolution of this specialized area of research suggests that sufficient research techniques and tools were garnered by the end of the third period.

Cluster #2—*Vinegar production.* The timespan associated with this cluster was 15 years (from 2000 to 2014). The main topic of this cluster was AAB. The five-year period from 2006 to 2010 was a highly active period with 39 references accounting for approximately 54% of the articles in the cluster. During this period, the succession, characterization, strain-typing, and genomic characteristics of AAB in vinegar fermentation were systematically studied [49,50,51]. AAB are the core microorganisms in vinegar production. These articles give us a better understanding of the involvement of AAB in vinegar fermentation. In the following period from 2011 to 2014, the main focus was on the application of eximious AAB and the optimization of fermentation conditions to improve vinegar production.

Cluster #5—*Bacterial diversity.* The mean year of this cluster was 2012. The main topic in the cluster is microbial community involvement in vinegar fermentation. Unlike pure culture fermentation, traditional vinegar fermentation requires a complex microflora. The function of the microbial community in both a temporal and spatial context is important when attempting to understand the mechanisms that underpin vinegar fermentation. In the start-up period from 2004 to 2008, research pertaining to this topic predominantly focused on the diversity of individual genus microorganisms in vinegar fermentation. In 2006 and 2008, Gullo et al. [52], De Vero et al. [53], and Solieri & Giudici [54] reported the diversity of AAB and yeasts involved in TBV fermentation. In the following period from 2009 until 2018, numerous articles investigating the microbial diversity during vinegar fermentation were continuously published. In 2011, the microbial community was monitored during acetic acid fermentation of Zhenjiang aromatic vinegar (ZAV) [55]. In 2012, microorganisms including yeasts, LAB, and AAB in the fermentation of SAV were isolated and characterized [56]. In 2013 and 2015, microbial succession and diversity during the fermentation of Tianjin Duliu mature vinegar and SAV were studied [57,58]. In 2016, research investigating microbial community composition during fermentation of Korean traditional black raspberry vinegar was also conducted [59].

Cluster #7—*Typical aroma.* This cluster is an old cluster, with 2005 representing the mean year. The core topic in the cluster is vinegar aroma. The period from 1999 to 2013 was relatively inactive. Most of the research focused on the profiling of various volatile compounds in vinegar using gas chromatography-mass spectrometry (GC-MS)/gas chromatography-olfactometry-mass spectrometry (GC-O-MS)/two-dimensional gas chromatography time-of-flight mass spectrometry (GC × GC-TOF-MS)-based chemical-analytical methodologies [60,61,62,63,64,65]. The two main extraction methods used for aroma compounds were solid-phase microextraction (SPME) [63,66] and stir bar sorptive extraction (SBSE) [60].

#### 3.3.2. Betweenness Centrality

Table 6 shows ten structural references in the co-citation network. These references are important bridges in connecting individual nodes or/and co-citation clusters in the whole network. These ten references can be considered as landmarks in the development of the vinegar research domain.

#### 3.3.3. Most Cited Articles

The most cited articles are usually deemed landmark articles owing to the significance of their contributions [13]. Table 7 lists the 10 most cited references, and 4 of them are reviews. The most cited reference in our dataset is a review article by Budak et al. [11] with 40 citations, followed by a journal article by Wu et al. [56] with 34 citations. The third most cited reference is a journal article by Xu et al. [55]. All three of these references were published in the last decade.

#### 3.3.4. Citation Bursts

Articles with citation bursts received particular attention from associated academic circles in a certain period. Strength and duration represent two important attributes of a citation burst. Table 8 lists the top 10 references with the strongest citation bursts from 1998 to 2019. This time interval is depicted as a blue line. The time period in which a reference was detected as having a burst is shown as a red line segment, indicating the initial and final year of the burst duration (the lines in Table 9 have the same meanings). Three of the ten references are review articles and two of these reviews have the strongest citation bursts. This result also indicates that the importance of review articles has been widely recognized by the academic community [82]. Integrated reviews may help researchers to keep abreast of knowledge-based developments and research fronts.

### 3.4. Theme Panels

Intuitively, the keywords from citing articles provide with a snapshot of the main topics in vinegar research. From the micro perspective, structurally essential (articles with high betweenness centrality) and inspirational references (articles with strong citation bursts) illustrate the landmarks and hotspots in this knowledge domain, and macroscopically, the co-citation network provides with an overview of the intellectual structure. By considering all three levels, related themes in the knowledge domain of vinegar research can predominantly be divided into six panels.

*Microorganisms*. Microorganisms have an important influence on the quality of final products. As previously discussed (see Section 3.3.1), the microbial diversity was extensively studied in many vinegars either by culture-dependent or culture-independent approaches. A better understanding of the microbial characteristics and ecology can help us to more optimally utilize eximious microorganisms while controlling the fermentation process in a more informed manner.

*Substances*. The substances in bulk vinegar originate from the following: (1) metabolites produced by microorganisms; (2) raw materials; and (3) chemical reactions between preexisting substances. At present, most of the research relating to substance analysis in vinegar can be categorized into two classes: flavor and functional substances. The detection and isolation of substances in vinegar is predominantly performed by chromatography, while data matrix processing is done using chemometrics. However, it should be noted that there are also some substances that contribute to both the flavor and the bioactivity of vinegar. For instance, ligustrazine not only has a roasted and caramel odor [91], but it also exhibits beneficial effects in relation to cardiovascular and cerebrovascular diseases and hypertension. In addition to antioxidant activity, phenolic compounds can also exhibit flavor properties [92].

*Health functions*. Antioxidant activity represents the main research focus in relation to the health benefits of vinegar. This is probably because of the large number of studies on polyphenols, which are well-recognized antioxidants. However, this may also be because of the availability of various well-established evaluation methods [93,94].

Other health functions including antiglycemic, hypolipidemic, antihypertensive, anti-obesity, antidiabetic, and immunoenhancement effects associated with vinegar were also extensively reported following both in vitro or/and in vivo evaluations. In addition, health effects associated with both individual components of vinegar and vinegar as a whole were studied, for example, by Johnston et al. [86], where the antiglycemic effects in adults due to vinegar intake were reported, and Fushimi et al. [70], where the hypolipidemic benefits of acetic acid were reported.

*Production technologies*. Owing to the low commercial value of most vinegars, technological innovation was often considered unprofitable [79]. However, with the development of a deeper understanding of microbial ecophysiology and fermentation mechanisms, technological innovations are constantly emerging. In addition to equipment upgrading, nowadays, technological innovations in vinegar production mainly focus on (i) the enhancement of industrial vinegar production by altering or optimizing operation modes [95,96,97], and (ii) microbial augmentation to improve vinegar production or quality [79,98,99].

*Adjuvant medicines*. This is a relatively insular research area compared with the major research areas in the knowledge domain. This topic involves the study of changes to substances and the efficacy of herbal medicines such as *Euphorbia kansui* [100,101,102] and *Radix bupleuri* [103,104] following fry-baking with vinegar.

*Vinegar residues*. Vinegar residues represent a small cluster in the knowledge domain (Cluster #12, Figure 5). This area of research concentrates on the extraction of vinegar residue. Representative articles from this cluster include Ran et al.’s article [105] on the separation of acetate from fresh vinegar residue and the application of hydrothermal treatment on washed vinegar residue to enhance methane production, Song et al.’s article [106] on the evaluation of nitrogen excretion in laying hens following feeding with waste vinegar residue, and Du et al.’s article [107] on the application of vinegar residue for the control of *Fusarium wilt* in cucumbers.

On the basis of the above analysis, it is apparent that microorganisms, substances, and health functions are three major themes in the knowledge domain. As shown in Figure 7, a logic model illustrates the relationship among the research themes currently studied in the vinegar research area.

### 3.5. Emerging Trends and Outlook

The evolution of vinegar-related research fronts can be evaluated by detecting the citation burst of literature in a certain period. Depending upon burst detection in the co-citation analysis, a total of 13 references was detected for the most recent bursts up until 2019 (Table 9). The Sigma metric, which measures both the structural centrality and citation burst of an article, shows that two review articles entitled *Functional properties of vinegar* and *Effect and Mechanisms of action of vinegar on glucose metabolism, lipid profile, and body weight* by Budak et al. [11] and Petsiou et al. [108], respectively, have the highest Sigma values. Thematically, all of the aforementioned 13 articles cover microorganisms, substances, and health functions.

The functions and interactions of microorganisms in dynamic spatiotemporal networks of microbial communities are critical for the rational regulation of vinegar fermentation. Up until now, both culture-dependent and -independent approaches have been widely used in the exploration of microbial ecology in complex microbial systems. Microbial culturomics has made the culture of difficult-to-culture microbes feasible [109]. Furthermore, integrated multi-omic and trans-omic analyses have provided us with insights into microbial diversity, gene expression, and microbial interaction, as well as allowed us to reconstruct global biochemical networks [110,111]. However, integration and visualization of the huge amount of omics data make it a formidable challenge to understand these data [112,113,114]. Unscrambling the omics data to provide a clear description of microbial ecology in vinegar fermentation will be an emerging focus in the future.

Before rationally regulating the fermentation process to forge a high-quality product, a valid evaluation criterion for the measurement of product quality should first be established. Besides, it is imperative that we are fully aware of the materials that we are consuming [119]. In general, the composition and content of various substances in a food matrix give appraisable indexes of the product quality. Indeed, many publications have reported the flavor substances in different vinegars. Nevertheless, no consensus has been reached on the characteristic flavor substances of a vinegar. One of the reasons for this ambiguity relates to differences between various methods. However, a more effective solution may encompass data processing using a uniform chemometric model to integrate data obtained by different methods. In terms of functional substances in vinegar, only a handful of substances have been analyzed. This is mainly because of the complicated nature of the vinegar matrix. An analogous example in this context is traditional Chinese medicine (TCM), which also involves a complex, open, and massive system [120]. Metabolomics plays an important role in comprehensive chemical profiling and quantitation of metabolites in TCM. As it stands, large-scale separation, detection, identification, and quantitation of metabolites in TCM have already been performed [121,122,123]. In addition, a polypharmacokinetics approach has been proposed to analyze potential functional substances in multicomponent natural products despite the diversity associated with the botanical chemical composition and complex effects elicited by associated metabolic pathways [124,125]. Dissection of the vinegar functional chemome is still in its infancy and is likely to become an emerging trend; a greater abundance of referential concepts and methods will most likely be adopted from TCM studies in the future.

Many associated health functions of vinegar have been verified following both in vitro or in vivo experiments. Nevertheless, most of the functional mechanisms have not yet been elucidated. This is because of limited knowledge relating to the functional chemome of vinegar, along with the diversity of both targets and metabolic pathways associated with functional ingredients. By tracking TCM studies, we found that the gut microbiota has emerged as a new and important research frontier when attempting to understand the function of TCM [126,127]. Because vinegar contains a myriad of compounds belonging to different chemical classes, this condiment will unavoidably interact with the gut microbiota after administration. However, to the best of our knowledge, there is still only a limited number of reports pertaining to interactions between vinegar and the gut microbiota, as well as the correlation between the gut microbiota and health functions of vinegar. Considering the pivotal role of the gut microbiota in human health, we contend that the effects of vinegar on the gut microbiota will draw more focused attention in future studies.

### 3.6. Evolutionary Stage of Vinegar Research

The sequential evolution of each scientific discipline follows a certain rule. According to Shneider’s theory [128], the development course of a scientific discipline exhibits four stages. At the first stage, scientists observe a previously known phenomenon or a new phenomenon and introduce new subject matter for the purposes of scientific research. Before conducting research, new theories, concepts, and hypotheses are proposed to establish and categorize research objects. The research object for vinegar research is the fermented food domain. To characterize research objects, analytical instruments or tools are developed or adopted from other fields. Next in the process is the second stage. During this stage, to identify the composition of vinegar, diverse instruments such as GC-MS, HPLC, and NIR are utilized. When associated tools are well prepared and widely used by researchers, then lots of data and results will be generated. The research then enters the third stage of the process. At this stage, more details will be uncovered, and a deeper understanding of a scientific problem is developed. Meanwhile, new questions may arise because of either the emergence of new phenomena or reconsideration of a problem from new perspectives. At the final stage, namely the fourth stage, new discoveries are seldom made. The main goal of this stage is to progress our knowledge in the area. This also constitutes a review stage, with the publication of comprehensive reviews and textbooks.

On the basis of the above analysis and discussion, we suggest that vinegar research is a specialized research area in the third stage of the aforementioned development course. Various analytical instruments are available and ready for use. Research issues are becoming more and more specific and a range of research results have enhanced our comprehension of this knowledge domain.

## 4. Conclusions

In conclusion, in this study, the vinegar research knowledge domain was visualized for the first time using CiteSpace. The dual-map overlay presented the disciplines involved in vinegar research in the context of a global map of science, and a co-citation network of the references sketched the intellectual structure of this domain. In addition, structural and inspirational articles were identified. Together, these analyses afford scientists with both a macroscopical understanding and a microscopical characterization of the knowledge domain as a whole. According to the research contents, vinegar research themes can be categorized into six panels: microorganisms, substances, health functions, production technologies, adjuvant medicines, and vinegar residues. By detecting the most recent burst articles up until 2019, emerging trends in vinegar research were discerned. It is believed that thorough analyses of omics data will be an emerging focus in microorganisms, while also representing a means of determining the spatiotemporal description of the microbial ecology in vinegar fermentation. This type of research will also encompass identification of the core flavor chemome of vinegar by using a uniform chemometric model to integrate multi-source data. Dissection of the vinegar functional chemome represents an emerging challenge, and it is hoped that helpful methodologies can be introduced from the TCM field. In terms of vinegar bioactivity research, the gut microbiota is likely to represent a new avenue for vinegar research. In addition, according to Shneider’s four-stage theory on scientific disciplines, vinegar research is a specialty research area in its third stage. Compared with traditional systematic reviews written by experts, this bibliometric analysis provides a timely, visual, and unbiased approach to track the development and explore the intellectual structure of specific knowledge domains.

The results of this study are based on the analysis of objective data. Though with stability and objectivity, there is a limitation associated with this study. In order to meet the data analysis requirements, we only used literature originally published in English from the WoSCC database. Therefore, the inclusion of papers published in other languages from other databases may increase the rigor of the study. Future studies looking at a greater number of databases (such as Scopus and patent databases) with broader coverage are strongly encouraged.

## Figures and Tables

**Figure 1 foods-09-00166-f001:**
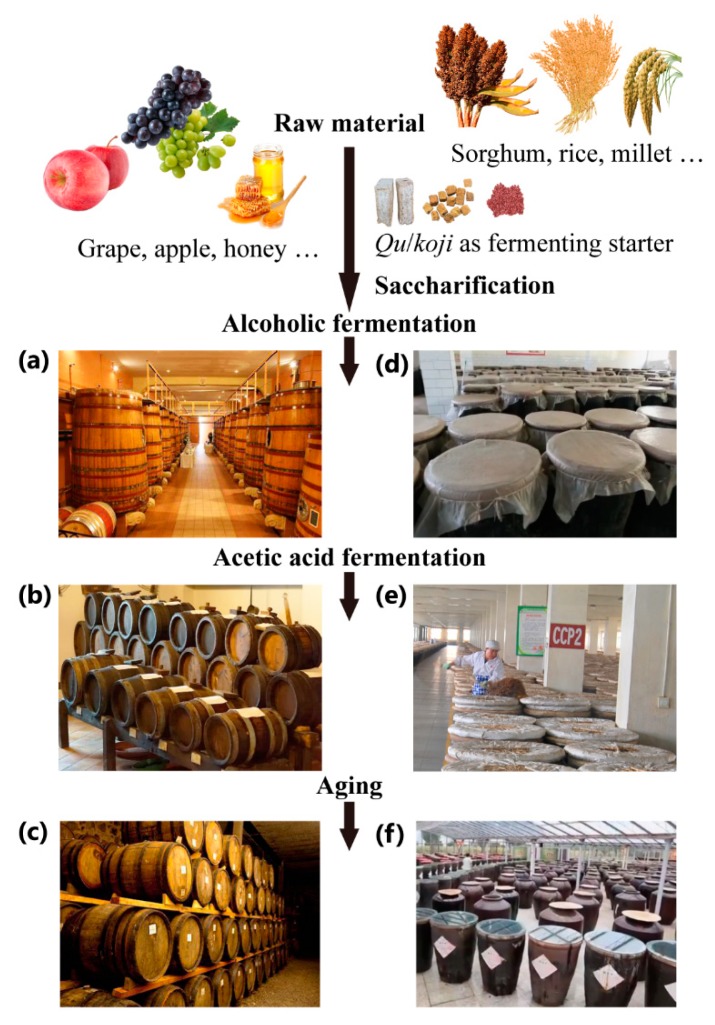
General production process for vinegar. (**a**) Alcoholic fermentation, (**b**) acetic acid fermentation, and (**c**) aging of grape vinegar in the barrel set (large special barrels for alcoholic fermentation and small ones for acetic fermentation and aging); (**d**) alcoholic fermentation of cereal vinegar in ceramic vats, (**e**) turning over the solid mash (called *Pei* in Chinese) during acetic acid fermentation of cereal vinegar, and (**f**) aging of cereal vinegar in the ceramic vats under sun.

**Figure 2 foods-09-00166-f002:**
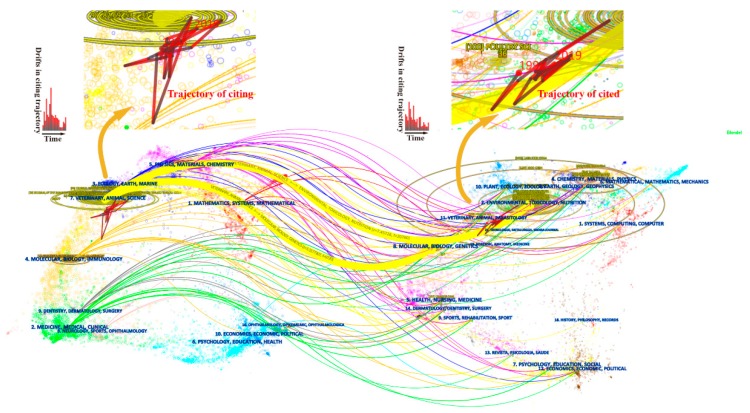
A dual-map overlay of the 883 publications on vinegar research.

**Figure 3 foods-09-00166-f003:**
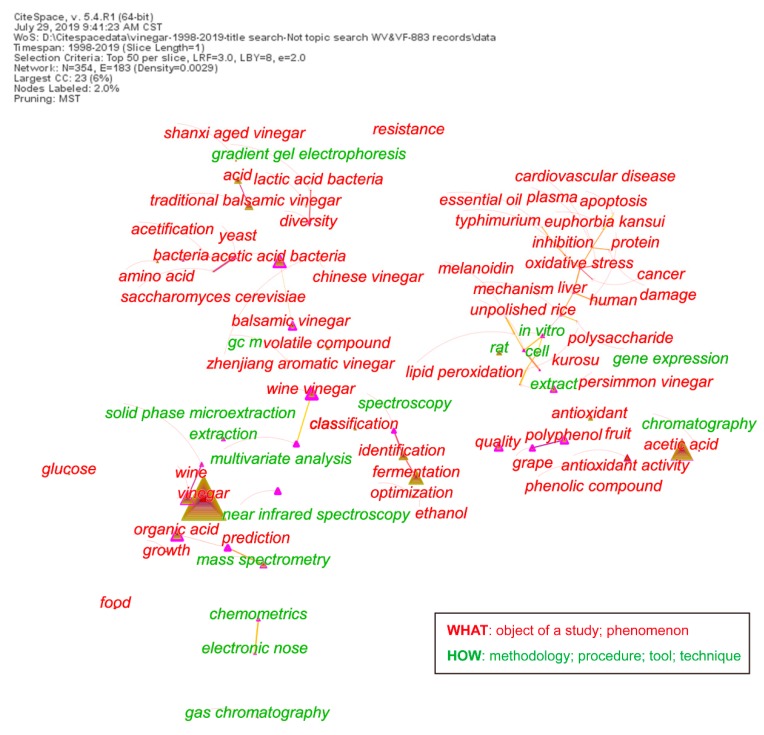
Minimum spanning tree of a keyword network based on articles published between 1998 and 2019.

**Figure 4 foods-09-00166-f004:**
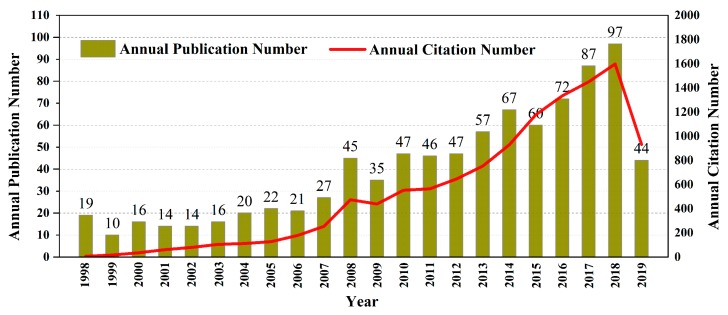
Annual publication and citation number from 1998 to 2019 at Web of Science (WoS).

**Figure 5 foods-09-00166-f005:**
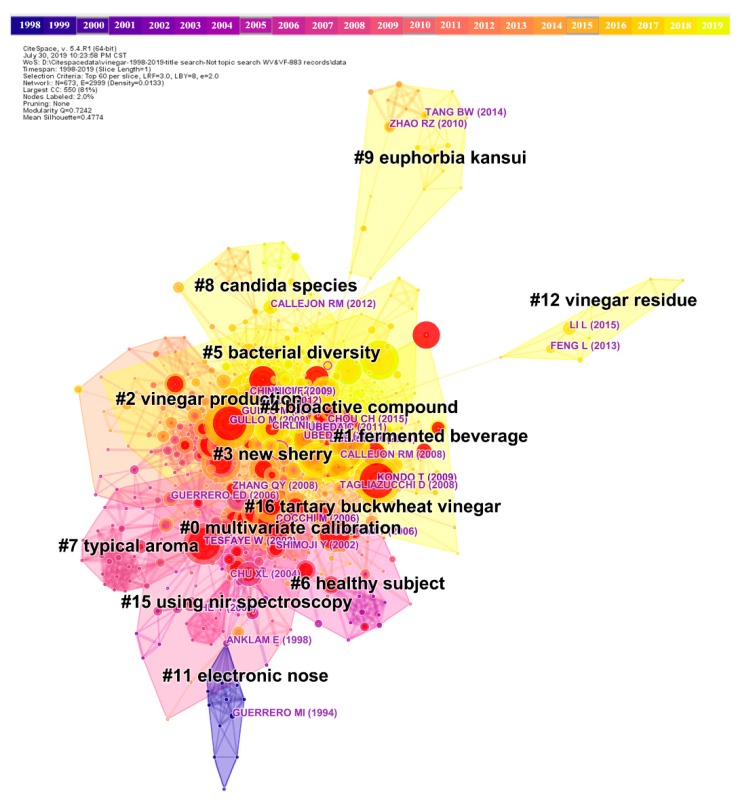
A landscape view of the co-citation network, generated by top 60 per slice between 1998 and 2019. All cluster labels were extracted from titles of citing articles using the log-likelihood ratio algorithm.

**Figure 6 foods-09-00166-f006:**
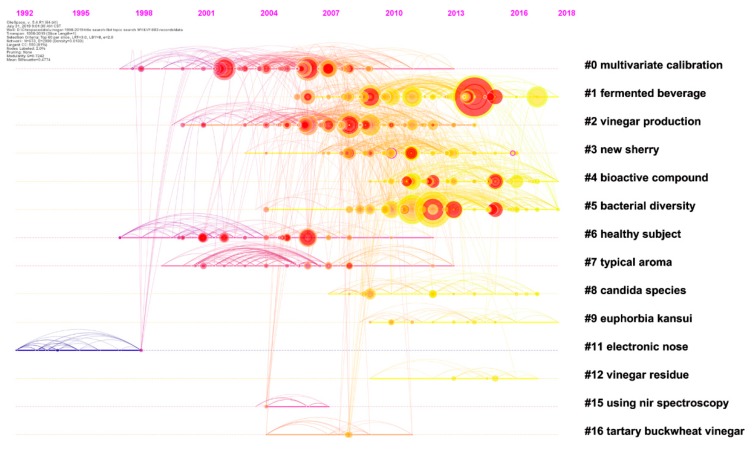
Timelines of co-citation clusters. Major clusters are labeled on the right.

**Figure 7 foods-09-00166-f007:**
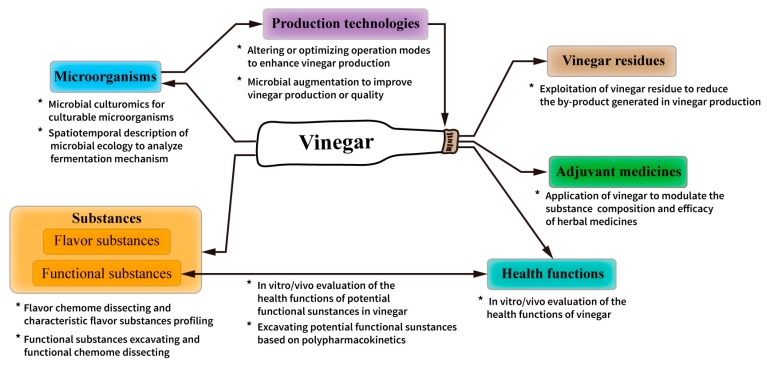
A logic model of related themes in current vinegar research areas.

**Table 1 foods-09-00166-t001:** Some representative vinegars in different countries. LSF, liquid-state fermentation; SSF, solid-state fermentation.

Geographical Distribution	Vinegar	Raw Material	Fermentation Technology
America	Malt vinegar	Barley, corn malt	LSF
Austria	Apple cider vinegar	Apple	LSF
Brazil	Alcohol vinegar	Alcohol	LSF
China	Shanxi aged vinegar (SAV)	Sorghum, barley, pea, bran, rice hull	SSF
China	Zhenjiang aromatic vinegar (ZAV)	Rice, glutinous rice, pea, wheat, barley, rice hull	SSF
China	Huixian persimmon vinegar	Persimmon, bran	SSF
England	Malt vinegar	Barley malt	LSF
France	Champagne wine vinegar	White grape	LSF
France	Red wine vinegar	Red grape	LSF
France	Walnut grape balsamic vinegar	Grape, walnut powder	LSF
Germany	Apple cider vinegar	Apple	LSF
Germany	White wine vinegar	White grape	LSF
Greece	Red wine vinegar	Grape, honey	LSF
Italy	Traditional balsamic vinegar (TBV)	Grape	LSF
Italy	White wine vinegar	White grape	LSF
Italy	Apple cider vinegar	Apple	LSF
Japan	Black rice vinegar	Brown rice	LSF
Japan	Kombucha vinegar	Kombucha tea	LSF
Malaysia	Nipa palm vinegar	Nipa palm sap	LSF
Mexico	Apple cider vinegar	Apple	LSF
New Zealand	Apple cider vinegar	Apple, honey	LSF
Portugal	Grape vinegar	Grape	LSF
Philippines	Coconut vinegar	Coconut sap	LSF
Philippines	Cane vinegar	Sugarcane juice	LSF
Spain	Sherry vinegar	Sherry wine	LSF
Spain	Sweet Moscatel vinegar	Sweet Moscatel wine	LSF
South Korea	Glutinous rice vinegar	Glutinous rice	LSF
Turkey	Grape vinegar	Grape	LSF
Turkey	Pomegranate vinegar	Pomegranate	LSF
Vietnam	Rice vinegar	Rice	LSF

**Table 2 foods-09-00166-t002:** Top 10 subject categories, countries, and institutions in terms of publications.

NO.	Name	Frequency	Percentage %
Subject categories			
1	Food Science & Technology	445	50.40
2	Chemistry	247	27.97
3	Chemistry, Applied	123	13.93
4	Biotechnology & Applied Microbiology	113	12.80
5	Agriculture	96	10.87
6	Nutrition & Dietetics	88	9.97
7	Chemistry, Analytical	84	9.51
8	Biochemistry & Molecular Biology	70	7.93
9	Microbiology	67	7.59
10	Agriculture, Multidisciplinary	55	6.23
Countries			
1	China	261	29.56
2	Japan	109	12.34
3	Spain	101	11.44
4	Italy	83	9.40
5	South Korea	72	8.15
6	USA	58	6.57
7	Turkey	31	3.51
8	Brazil	29	3.28
9	Malaysia	19	2.15
10	Germany	17	1.93
Institutions			
1	University of Modena and Reggio Emilia (Italy)	35	3.96
2	University of Seville (Spain)	27	3.05
3	University of Cadiz (Spain)	25	2.83
4	Jiangnan University (China)	19	2.15
5	Jiangsu University (China)	18	2.04
6	Chinese Academy of Sciences (China)	17	1.92
7	University of Rovira i Virgili (Spain)	15	1.70
8	Tianjin University of Science and Technology (China)	13	1.47
9	China Agricultural University (China)	13	1.47
10	Spanish National Research Council (CSIC, Spain)	13	1.47

**Table 3 foods-09-00166-t003:** Top 10 journals that published articles on vinegar research from 1998 to 2019.

NO.	Journal Name	JCR Category ^a^	Rank in Category	Quartile in Category	IF	Eigenfactor Score	Average JIF Percentile	Frequency	Percentage %
1	*Food Chemistry*	Chemistry, Applied	5/71	Q1	5.399	0.10387	92.600	39	4.41
2	*Journal of Agricultural and Food Chemistry*	Agriculture, Multidisciplinary	3/56	Q1	3.571	0.06656	85.384	34	3.85
3	*Food Science and Biotechnology*	Food Science & Technology	108/135	Q4	0.888	0.00355	20.370	19	2.15
4	*Journal of the Japanese Society for Food Science and Technology-Nippon Shokuhin Kagaku Kogaku Kaishi*	Food Science & Technology	129/135	Q4	0.262	0.00023	4.815	18	2.04
5	*Journal of the Science of Food and Agriculture*	Agriculture, Multidisciplinary	9/56	Q1	2.422	0.01875	73.883	18	2.04
6	*European Food Research and Technology*	Food Science & Technology	58/135	Q2	2.056	0.00607	57.407	17	1.92
7	*International Journal of Food Microbiology*	Food Science & Technology	16/135	Q1	4.006	0.01961	79.786	17	1.92
8	*Journal of Food Engineering*	Engineering, Chemical	28/138	Q1	3.625	0.01831	80.592	16	1.81
9	*Food Analytical Methods*	Food Science & Technology	45/135	Q2	2.413	0.00854	67.037	14	1.58
10	*Food Control*	Food Science & Technology	11/135	Q1	4.248	0.02984	92.222	14	1.58

Note: all journal ranks, quartiles, impact factors (IF), eigenfactor scores, and average journal impact factor (JIF) percentiles as of 2018 according to journal citation reports (JCRs). ^a^ JCR category corresponds to the first category listed in the JCR.

**Table 4 foods-09-00166-t004:** Temporal major clusters of co-cited references.

Cluster #	Size	Silhouette	Label (LSI ^a^)	Label (LLR ^a^)	Label (MI ^a^)	Mean (Year)
0	83	0.737	Modena	Multivariate calibration	Powerful combination	2004
1	75	0.692	Vinegar	Fermented beverage	Pomegranate vinegar	2012
2	73	0.810	Acetic acid bacteria	Vinegar production	Pomegranate vinegar	2007
3	57	0.602	Maceration	New sherry	Oat vinegar	2010
4	53	0.623	Vinegar	Bioactive compound	Aroma constituent	2014
5	43	0.842	Vinegar	Bacterial diversity	Benchmarking laboratory-scale pomegranate vinegar	2012
6	36	0.945	Vinegar	Healthy subject	Vinegar intake	2003
7	36	0.868	Sorptive extraction	Typical aroma	Concentrated fruit vinegar	2005
8	33	0.911	Protected designation	Candida species	Spanish wine vinegar	2011
9	21	0.955	Vinegar	*Euphorbia kansui*	Cell membrane constituent	2013
11	13	0.988	Characterization	Electronic nose	Using gas chromatography	1994
12	11	0.991	Vinegar residue	Vinegar residue	Using vinegar residue biochar	2014

**^a^** Label algorithm, LSI: latent semantic indexing; LLR: log-likelihood ratio; MI: mutual information.

**Table 5 foods-09-00166-t005:** Most cited and highest citation coverage articles in typical clusters.

Cluter #	Cited References		Citing Articles	
	Cites	Cited References	Coverage %	First Author (Year) Title
1 & 6	40	[11]	27	Li, S. (2015) Microbial diversity and their roles in the vinegar fermentation process
	24	[67]	9	Lynch, K. M. (2019) Physiology of acetic acid bacteria and their role in vinegar and **fermented beverages**
	21	[68]	8	Chen, H. Y. (2016) Vinegar functions on health: constituents, sources, and formation mechanisms
	20	[69]	11	Leeman, M. (2005) Vinegar dressing and cold storage of potatoes lowers postprandial glycaemic and insulinaemic responses in **healthy subjects**
	19	[70]	10	Ostman, E. (2005) Vinegar supplementation lowers glucose and insulin responses and increases satiety after a bread meal in **healthy subjects**
	18	[71]	5	Johnston, C. S. (2005) Vinegar and peanut products as complementary foods to reduce postprandial glycemia
3 & 4	18	[35]	24	Li, S. (2015) Microbial diversity and their roles in the vinegar fermentation process
	18	[72]	5	Aykin, E. (2015) Bioactive components of mother vinegar
	17	[73]	4	Cejudo-Bastante, M. J. (2013) Study of the volatile composition and sensory characteristics of **new sherry** vinegar-derived products by maceration with fruits
	16	[74]	9	Kawa-Rygielska, J. (2018) Bioactive compounds in cornelian cherry vinegars
	15	[75]	8	Xia, T. (2018) Evaluation of nutritional compositions, **bioactive compounds**, and antioxidant activities of Shanxi aged vinegars during the aging process
	14	[76]	8	Xia, T. (2018) Shanxi aged vinegar prevents alcoholic liver injury by inhibiting CYP2E1 and NADPH oxidase activities
0	24	[45]	12	Liu, F. (2011) Variety identification of rice vinegars using visible and near infrared spectroscopy and **multivariate calibrations**
	22	[77]	10	Liu, F. (2011) Detection of organic acids and pH of fruit vinegars using near-infrared spectroscopy and **multivariate calibration**
	18	[47]	8	Chen, Q. S. (2012) Simultaneous measurement of total acid content and soluble salt-free solids content in Chinese vinegar using near-infrared spectroscopy
2	26	[78]	19	Fernandez-Perez, R. (2010) Rapid molecular methods for enumeration and taxonomical identification of acetic acid bacteria responsible for submerged **vinegar production**
	21	[79]	16	Fernandez-Perez, R. (2010) Strain typing of acetic acid bacteria responsible for **vinegar production** by the submerged elaboration method
	20	[52]	12	Torija, M. J. (2010) Identification and quantification of acetic acid bacteria in wine and vinegar by TaqMan-MGB probes
5	34	[56]	7	Nie, Z. Q. (2017) Unraveling the correlation between microbiota succession and metabolite changes in traditional Shanxi aged vinegar
	29	[55]	6	Gan, X. (2017) Diversity and dynamics stability of bacterial community in traditional solid-state fermentation of Qishan vinegar
	19	[57]	5	Milanovic, V. (2018) Profiling white wine seed vinegar **bacterial diversity** through viable counting, metagenomic sequencing and PCR-DGGE
7	8	[60]	17	Callejon, R. M. (2008) Defining the **typical aroma** of sherry vinegar: sensory and chemical approach
	8	[80]	17	Callejon, R. M. (2008) Targeting key aromatic substances on the **typical aroma** of sherry vinegar
	8	[81]	9	Callejon, R. M. (2008) Optimization and validation of headspace sorptive extraction for the analysis of volatile compounds in wine vinegars

Note: cluster labels occurring in citing article titles are in bold. The six most cited and highest citation coverage articles are listed in integrated clusters.

**Table 6 foods-09-00166-t006:** Cited references with the top 10 highest betweenness centrality.

References	Centrality	Document Type	Cluster #	Article Contents
[83]	0.13	Journal article	3	Derived sherry wine vinegar was obtained by maceration with fruits; polyphenolic content and antioxidant activity were determined.
[84]	0.11	Journal article	3	The development of an orange-based vinegar; the polyphenolic and volatile content were determined.
[45]	0.09	Journal article	0	A gas chromatographic method to determine the sugars and organic acids in vinegar was developed; a chemometric technique (Tucker 3) was applied in data analysis.
[77]	0.08	Review	0	Wine vinegar processing technology (including bacterial strain, acetification system design optimum conditions), authentication, and quality evaluation were reviewed.
[52]	0.07	Journal article	2	Isolation and characterization of acetic acid bacteria (AAB) from TBV.
[78]	0.07	Review	2	Phenotypic traits of AAB in TBV production, TBV defects, and selection criteria for AAB starter culture were reviewed.
[85]	0.07	Journal article	0	The phenolic content and antioxidant activity of high-molecular-weight melanoidins in ZAV were determined.
[47]	0.07	Journal article	0	An ^1^H NMR method to simultaneously determine the main organic components of vinegars was developed.
[60]	0.06	Journal article	7	A new technique (SBSE) for extracting volatile compounds of vinegar was introduced, and a comparison of this method with solid-phase microextraction (SPME) method was made.
[86]	0.06	Journal article	1	The antiglycemic properties of vinegar in adults were evaluated.

**Table 7 foods-09-00166-t007:** Top 10 most cited references.

References	Citation Counts	Document Type	Cluster #	Article Contents
[11]	40	Review	1	The health effects of vinegar were reviewed.
[56]	34	Journal article	5	Microorganisms including yeasts, lactic acid bacteria (LAB), and AAB were isolated and characterized based on phenotypic and genotypic approaches.
[55]	29	Journal article	5	Denaturing gradient gel electrophoresis combined with clone library was used to analyze the microbial diversity during the fermentation process of ZAV.
[78]	26	Review	2	Phenotypic traits of AAB in TBV production, TBV defects, and selection criteria for AAB starter culture were reviewed.
[45]	24	Journal article	0	A gas chromatographic method to determine the sugars and organic acids in vinegar was developed; a chemometric technique (Tucker 3) was applied in data analysis.
[67]	24	Journal article	1	The antiobesity effect of vinegar in adults was evaluated.
[77]	22	Review	0	Wine vinegar processing technology (including bacterial strain, acetification system design optimum conditions), authentication, and quality evaluation were reviewed.
[79]	21	Journal article	2	The application of selected Acetobacter pasteurianus strains for TBV production was assessed, and its persistence and species succession were evaluated.
[68]	21	Journal article	1	The hypolipidemic effect of apple cider vinegars produced with and without inclusion of maceration was evaluated in high-cholesterol-fed rats.
[69]	20	Review	1	Varieties, production, volatile compounds, organic acids, bioactive compounds, and health benefits of vinegar were reviewed.

**Table 8 foods-09-00166-t008:** Top 10 references with the strongest citation bursts.

References	Strength of Burst	Document Type	Duration of Burst	Cluster #	Article Contents
[77]	10.60	Review	▂▂▂▂ ▂▂ ▃▃▃▃▃▃▃ ▂▂▂▂▂▂▂▂▂	0	Wine vinegar processing technology (including bacterial strain, acetification system design optimum conditions), authentication, and quality evaluation were reviewed.
			2004–2010		
[11]	10.25	Review	▂▂▂▂▂▂▂▂▂▂▂▂▂▂▂▂ ▂▂ ▃▃▃▃	1	The health effects of vinegar were reviewed.
			2016–2019		
[70]	7.43	Journal article	▂▂▂▂▂▂▂▂ ▂▂▂ ▃▃▃▃▃▃ ▂▂▂▂▂	6	The prevention of hyperlipidemia by dietary acetic acid in high-cholesterol fed rats was evaluated.
			2009–2014		
[52]	7.30	Journal article	▂▂▂▂▂▂▂▂ ▂▂ ▃▃▃ ▂▂▂▂▂▂▂▂▂	2	Isolation and characterization of AAB from TBV.
			2008–2010		
[43]	7.18	Journal article	▂▂▂▂ ▂▂ ▃▃▃▃▃ ▂▂▂▂▂▂▂▂▂▂▂	0	The efficiency of high performance liquid chromatography (HPLC) and GC combined with a solid-phase extraction method with C18 and NH(2) exchangers to determine the carboxylic acids was evaluated.
			2004–2008		
[87]	6.89	Journal article	▂▂▂▂▂▂▂▂ ▂ ▃▃▃▃▃▃▃▃ ▂▂▂▂▂	0	A near-infrared spectroscopy method was developed to quantify the degree of adulteration of wheat flours.
			2007–2014		
[48]	6.66	Journal article	▂▂▂▂▂▂▂▂ ▂▂▂ ▃▃▃▃ ▂▂▂▂▂▂▂	0	A near-infrared spectroscopy method for quality control of wine vinegar through the determination of 14 parameters was developed.
			2009–2012		
[88]	6.21	Review	▂▂▂▂▂▂▂▂ ▂▂▂▂ ▃▃▃▃▃ ▂▂▂▂▂	1	Scientific evidences for medicinal uses of vinegar, especially as an antiglycemic agent, were reviewed.
			2010–2014		
[89]	6.05	Journal article	▂▂▂▂▂▂▂▂▂▂▂▂▂▂▂▂▂ ▂▂ ▃▃▃	1	Antioxidant activity, antimicrobial, mineral, volatile profiles, and microbiota of 20 traditional Turkey vinegars were characterized.
			2017–2019		
[90]	5.93	Journal article	▂▂▂ ▂▂▂▂ ▃▃▃▃▃ ▂▂▂▂▂▂▂▂▂▂	6	The antihypertensive effect of acetic acid and vinegar on spontaneously hypertensive rats was evaluated.
			2005–2009		

**Table 9 foods-09-00166-t009:** References with the most recent bursts until 2019.

References	Centrality	Strength of Burst	Sigma	Document Type	Duration of Burst	Cluster #	Theme Panels
[57]	0.01	3.33	1.05	Journal article	▂▂▂▂▂▂▂▂▂▂▂▂▂▂▂ ▂▂ ▃▃▃▃▃	5	Microorganisms
					2015–2019		
[11]	0.05	10.25	1.66	Review	▂▂▂▂▂▂▂▂▂▂▂▂▂▂▂▂ ▂▂ ▃▃▃▃	1	Health functions
					2016–2019		
[35]	0.01	5.19	1.05	Journal article	▂▂▂▂▂▂▂▂▂▂▂▂▂▂▂▂▂ ▂ ▃▃▃▃	4	Substances, health functions
					2016–2019		
[58]	0.01	4.68	1.03	Journal article	▂▂▂▂▂▂▂▂▂▂▂▂▂▂▂▂▂ ▂ ▃▃▃▃	5	Microorganisms
					2016–2019		
[108]	0.02	4.03	1.10	Review	▂▂▂▂▂▂▂▂▂▂▂▂▂▂▂▂ ▂▂ ▃▃▃▃	1	Health functions
					2016–2019		
[64]	0.02	3.34	1.08	Journal article	▂▂▂▂▂▂▂▂▂▂▂▂▂ ▂▂▂▂▂ ▃▃▃▃	5	Substances
					2016–2019		
[74]	0.01	3.21	1.02	Journal article	▂▂▂▂▂▂▂▂▂▂▂▂▂ ▂▂▂▂▂ ▃▃▃▃	3	Substances
					2016–2019		
[89]	0.00	6.05	1.00	Journal article	▂▂▂▂▂▂▂▂▂▂▂▂▂▂▂▂▂ ▂▂ ▃▃▃	1	Substances, microorganisms, health functions
					2017–2019		
[115]	0.01	5.18	1.04	Journal article	▂▂▂▂▂▂▂▂▂▂▂▂▂▂ ▂▂▂▂▂ ▃▃▃	4	Substances
					2017–2019		
[31]	0.00	4.31	1.01	Journal article	▂▂▂▂▂▂▂▂▂▂▂▂▂ ▂▂▂▂▂▂ ▃▃▃	4	Substances
					2017–2019		
[116]	0.00	4.31	1.01	Journal article	▂▂▂▂▂▂▂▂▂▂▂▂▂▂▂▂▂ ▂▂ ▃▃▃	5	Substances, microorganisms
					2017–2019		
[117]	0.00	3.46	1.00	Journal article	▂▂▂▂▂▂▂▂▂▂▂▂▂▂▂▂ ▂▂▂ ▃▃▃	1	Health functions
					2017–2019		
[118]	0.00	3.45	1.01	Journal article	▂▂▂▂▂▂▂▂▂▂▂▂▂▂▂▂▂ ▂▂ ▃▃▃	1	Health functions
					2017–2019		

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
