# Peer review of "Knowledge Domain and Emerging Trends in Vinegar Research: A Bibliometric Review of the Literature from WoSCC"

_foods, 2020, doi:10.3390/foods9020166_

Round 1
Reviewer 1 Report
Dear authors,
The paper shows clearly the knowledge map of vinegar research. The scope and introduction properly defines the state of art, and results are presented in an excellent way. However, there are several issues that should be addressed before publication:
Why is only included documents of WoSCC and not other databases such as Scopus? Figure 2 is confussing. Its explanation should be improved. There is no discussion of results with previous researches. Finally, it has been detected 47% of plagiarism by Turnitin. I attach the results file.
Reviewer 2 Report
Dear authors,
Please, revise the following parts of the manuscript as these parts are too long and the main thread is lost.
1. Introduction should be shortened.
2. Sub-section 3.3.1. should be shortened.
2. Sub-section 3.4. should be shortened. The figure presented here is very nice and presents a lot of information given in the text.
3. Sub-section 3.5 should be shortened.
In addition, some Figures are not needed, such as Figure 2 (it is even not readable), so it can be omnited or insert as supporting materials. The same with Figures 3, 6, 7.
Reviewer 3 Report
This manuscript makes a bibliometric analysis that materializes the state of the art about vinegar in scientific literature in a clear, organized and detailed way. It specifically addresses the main findings of the data analysis and justifies them correctly.
The abstract exposes with simple and clear style a small justification and above all the objective the method followed and a synthesis of the conclusions.
The state of the art is current and complete. Likewise, it is not limited to descriptive aspects, but introduces important elements of criticism, justified and documented.
The title is representative and concisely groups the content of the study. However, it must contain the database in which the study (Web of Science) was conducted to avoid a biased interpretation of the reader.
On the other hand, it is important that the manuscript contains a greater specification of methodological limitations: What were the obstacles encountered during the data collection process? Why have some elements been removed within the bibliographic search? What are the inclusion and exclusion criteria, and what is the theoretical basis for that decision? What has been the justification for the election of the first period with respect to others?
Finally, it would be convenient to specify in a more concrete way what is the prospective study, why is research necessary in this area? Why are the results found during the investigation valuable?
In short, the manuscript has great quality, but the revision of some aspects is recommended to improve its content.
Round 2
Reviewer 1 Report
Dear authors,
The reviewed version of the paper has addressed every issue I pointed out last time. Furthermore, plagiarism is no longer detected.
Best regards.